# Plant Ribonuclease J: An Essential Player in Maintaining Chloroplast RNA Quality Control for Gene Expression

**DOI:** 10.3390/plants9030334

**Published:** 2020-03-05

**Authors:** Amber M. Hotto, David B. Stern, Gadi Schuster

**Affiliations:** 1Boyce Thompson Institute, Ithaca, NY 14853-1801, USA; amh264@cornell.edu (A.M.H.); ds28@cornell.edu (D.B.S.); 2Faculty of Biology, Technion-Israel Institute of Technology, Haifa 32000, Israel

**Keywords:** RNA degradation, antisense RNA, β-CASP proteins, RNA processing, ribonuclease J

## Abstract

RNA quality control is an indispensable but poorly understood process that enables organisms to distinguish functional RNAs from nonfunctional or inhibitory ones. In chloroplasts, whose gene expression activities are required for photosynthesis, retrograde signaling, and plant development, RNA quality control is of paramount importance, as transcription is relatively unregulated. The functional RNA population is distilled from this initial transcriptome by a combination of RNA-binding proteins and ribonucleases. One of the key enzymes is RNase J, a 5′→3′ exoribonuclease and an endoribonuclease that has been shown to trim 5′ RNA termini and eliminate deleterious antisense RNA. In the absence of RNase J, embryo development cannot be completed. Land plant RNase J contains a highly conserved C-terminal domain that is found in GT-1 DNA-binding transcription factors and is not present in its bacterial, archaeal, and algal counterparts. The GT-1 domain may confer specificity through DNA and/or RNA binding and/or protein–protein interactions and thus be an element in the mechanisms that identify target transcripts among diverse RNA populations. Further understanding of chloroplast RNA quality control relies on discovering how RNase J is regulated and how its specificity is imparted.

“RNA quality control” is a term that emerged some 20 years ago to describe pathways that recognize and destroy aberrant RNAs generated through processing or translational abnormalities or chemical damage [1,2,3,4]. Another class of transcripts subjected to quality control are non-coding RNAs (ncRNAs), particularly the subset that comprises antisense RNAs (asRNAs). Sense–antisense base pairing can be essential to RNA function, for example in RNA interference or the regulation of sense RNA translation, in both prokaryotic and eukaryotic regulatory pathways [5,6,7,8]. Overexpression of asRNAs, however, can undesirably inhibit sense RNA function, as occurs in chloroplasts when ribonuclease (RNase) J is downregulated [9]. This raises the question of how cells distinguish functional from deleterious asRNAs.

The chloroplast is an excellent system in which to probe mechanisms of RNA quality control, including asRNAs. Chloroplasts have compact and well-characterized genomes with attributes of the bacterial systems from their endosymbiotic progenitors, overlaid by regulatory functions required in their semi-autonomous eukaryotic environment. One of these regulatory functions is represented by RNase J, the topic of this mini-review, which appears to have an essential role in surveilling the asRNA population, in addition to maturing RNA 5′ ends [10]. The importance of this role derives from the relaxed transcription found in chloroplasts, as described in more detail below. More generally, a complete understanding of RNase J functions is essential to complete the picture of chloroplast gene expression mechanisms.

## 1. An Overview of Chloroplast Gene Expression

Chloroplast gene expression has been recently reviewed, including transcription initiation [11] and termination [12], RNA splicing [13] and editing [14], and translation [15]. Briefly, in plants, a two-polymerase system transcribes both strands of the −150 kb plastid genome from a large number of transcription start sites (TSS), numbering over 200 in barley [16] and *Arabidopsis* [17]. Inefficient transcription termination necessitates massive post-transcriptional processing to shape the accumulating transcriptome. This processing is mediated by an array of RNases and accessory factors.

The major chloroplast endoribonucleases in *Arabidopsis* are RNase E [18,19] and RNase J. RNase E appears to be involved in cleaving primary transcripts and works in concert with the RNA-binding protein (RBP) and transcriptional regulator RHON1 [20,21]. The major chloroplast exoribonucleases are polynucleotide phosphorylase (PNPase) and RNase II/RNR1. Both PNPase and RNase II trim RNA in the 3′→5′ direction and are inhibited by stem–loop structures and RBPs. They have been shown to act on a wide variety of mRNA, tRNA, and rRNA substrates and often operate cooperatively [22]. The RBPs that inhibit or guide RNases are key players in shaping the chloroplast transcriptome and mainly belong to sequence-specific helical repeat protein families. In land plants, the dominant family is pentatricopeptide repeat (PPR) proteins [23], whereas in *Chlamydomonas* the main family is octotricopeptide repeat (OPR) proteins ([24,25] and references therein). The tight and specific binding of RBPs yields small RNA (sRNA) footprints that have been catalogued at a genomic level for both *Arabidopsis* and *Chlamydomonas* [26,27].

## 2. Relaxed Transcription in Chloroplasts

Chloroplast RNA quality control is made necessary by relaxed transcription initiation and inefficient transcription termination. For example, a study of dozens of diverse plastid genomes, some of them quite large, showed that ≥85% of genome sequences are present in RNA-Seq libraries [28]. The largest plastid genome reported, that of *Haematococcus lacustris* at 1352 kb, rife with repeated elements, is also nearly fully transcribed [29]. The incipient RNA population must have distinguishing characteristics that lead to widely varying levels of accumulating transcripts.

The transcriptional landscape of *Arabidopsis* chloroplasts has been studied in most detail. Using RNA-Seq, it was found that apart from the well-characterized genic transcripts, *Arabidopsis* chloroplasts possess >100 ncRNAs, many of which are antisense to known genes, and that read coverage extends throughout the genome [30]. Some of the ncRNAs are abundant, lengthy transcripts that were previously unknown, and some contain open reading frames. The function(s) of most ncRNAs remains untested, although evidence suggests that certain asRNAs regulate gene expression or RNA processing [31,32].

As the chloroplast is neither fully eukaryotic nor prokaryotic, complete analysis of chloroplast transcripts from RNA-Seq data requires specialized methods. Most RNA-Seq library protocols and analytical pipelines were designed to analyze polyadenylated nucleus-encoded RNA, rendering them unusable for chloroplast transcriptome analysis, where polyadenylation is rare and RNAs often overlap and may be post-transcriptionally edited. This difficulty was overcome by the development of strand-specific protocols and appropriate bioinformatic pipelines, such as ChloroSeq [33,34,35]. Most recently, specialized RNA-Seq libraries were created to capture chloroplast RNA 5′ and 3′ termini, a process called Terminome-seq [17]. The relevance of RNA-Seq and Terminome-seq to RNA quality control is that they allow a global examination of transcript abundance and termini, respectively, in mutants or under other conditions where quality control is compromised [17].

## 3. Ribonuclease J and β-CASP Proteins

RNase J1 and the related J2 were first described in *B. subtilis* [36]. RNase J-CPSF (cleavage and polyadenylation specificity factor) homologs are present in most bacteria, Archaea, chloroplasts, and eukaryotic cells (Figure 1), suggestive of a ribonuclease that appeared early in evolution [37,38,39,40]. These proteins belong to a large group denoted “β-CASP”, of which a subgroup of β-CASP ribonucleases harbors dual endo- and 5′→3′ exoribonucleolytic activities. The other β-CASP proteins are involved in DNA repair and recombination as well as other functions [41,42]. In archaea, the β-CASP ribonuclease subgroup has been further divided into three major groups, two with defined orthologues of the eukaryotic CPSF-73 (see description below) and therefore designated CPSF types, and the other orthologous to bacterial RNase J and therefore designated RNase J type [39,40,43], which includes chloroplast RNase J. The domain structure, length, amino acid sequences, and catalytic mechanism of RNase J and related CPSF proteins are mostly conserved. These proteins contain the seven signature motifs of the metallo-β-lactamase (MBL) and β-CASP domains, I (D), II (HxHxDA), III (H), IV (D), A (D/G), B (H), and C (H) (Figure 1), that together participate in the coordination of two catalytic Zn^2+^ ions [44,45]. RNase J is active as a dimer or a tetramer, and the amino acid sequence responsible for oligomerization is located at the C-terminus. Plant RNase Js contain, in addition to the MBL-β-CASP motifs, a chloroplast transit peptide at the N-terminus and a conserved GT-1 domain that was previously identified in transcription factors at the C-terminus (discussed below).

Crystal structures of bacterial and archaeal RNase J predict a combination of 5′→3′ exonuclease and endonuclease activities, both of which have been observed biochemically in vitro, with the exonuclease activity being dependent on the 5′ end phosphorylation state [44,45,46,47,48]. Most RNase Js display both 5′→3′ exonucleolytic and endonucleolytic activities when tested in vitro. *Chlamydomonas reinhardtii* RNase J (*Cr*RNase J) is one of only three family members reported to exhibit exclusively endonucleolytic activity in vitro [37,49,50]. On the other hand, *Bacillus subtilis* RNase J1 is mainly exonucleolytic in vitro [51], whereas *Arabidopsis* RNase J (*At*RNase J) displays robust endonucleolytic and relatively minor exonucleolytic activities in vitro [52]. The biological significance and structural basis of the variable exo- and endonucleolytic activities are unknown, but one can predict substrate preferences. For example, exonucleolytic activity might target chemically suitable 5′ RNA termini more efficiently, while an endonuclease could catalyze internal processing or early steps of RNA degradation. In addition to the nature of the RNA 5′ end, the structure of the RNA, as well as other proteins involved, could affect the type of activity carried out by RNase J.

The most well studied eukaryotic member of this group is a cleavage and polyadenylation specificity factor of 73 kDa (CPSF-73). This protein is the endonuclease component of a multi-protein complex that plays a key role in pre-mRNA 3’-end formation. It cleaves at a CA motif 20–30 nt downstream of an AAUAAA polyadenylation consensus sequence and interacts with poly(A) polymerase and other factors to bring about cleavage and polyadenylation of pre-mRNAs in mammalian cells [38,53,54,55]. In addition, it functions as a 5′→3′ exoribonuclease in the maturation of histone pre-mRNA [56]. Most archaea encode one or several RNase J/β-CASP homologous proteins and either RNase R or the archaeal exosome. In the group of methanogenic archaea, genes encoding RNase R or the archaeal exosome are not present, suggesting the possibility that RNA processing and degradation are carried out exclusively by RNase J-CPSF proteins [39,49]. RNase J is present in many, but not all bacteria, and those that do not have it, like *Escherichia coli*, contain the other major endoribonuclease, RNase E. Cyanobacteria that are closely related to the evolutionary ancestor of plant chloroplasts contain both RNase J and RNase E, as do plant chloroplasts, with the exception of the green alga *Chlamydomonas reinhardtii*, which possesses only RNase J.

Endonuclease activity has not been identified so far in the degradation of mitochondrial transcripts. However, RNase Z (ELAC2), which is a CPSF homologue, is a mitochondrial endoribonuclease that processes the 3′ end of tRNA precursors. LACTB2 is an endoribonuclease that is present in human mitochondria, belongs to the MBL protein super family, and is possibly involved in RNA quality control [57]. RNase P, which is responsible for the 5′-end processing of tRNAs, is an additional mitochondrial endoribonuclease.

## 4. The Plant RNase J GT-1 Domain

In spite of their overall conservation with bacterial, archaeal, and animal RNase J-CPSF members, plant RNase Js are distinguished by a C-terminal extension with high homology to the GT-1 DNA-binding domain (Figure 1) [9,52]. The GT-1 domain was defined initially in pea and subsequently in about 30-member families of *Arabidopsis*, wheat, and rice transcription factors that regulate various developmental processes and are stress-responsive [58,59,60,61]. The DNA-binding domain of GT factors features a trihelix structure, which contains three conserved tryptophan residues and an amphipathic helix (Figure 2). The GT-1 domain recognizes a degenerate core sequence of 5′-G–Pu–(T/A)–A–A–(T/A)-3′, called the GT element. Such AU-rich sequences are common in intergenic regions of the chloroplast genome.

In order to examine the conservation degree of the GT-1 domain in plant RNase J, its predicted structure was superimposed on that of the known GT-1 transcription factor PDB 2EBI (Figure 2). The DNA–GT-1 interface was located exactly as predicted by the conserved, electropositive, tryptophan-rich interface [52,62]. The predicted structure also displayed similar physicochemical characteristics and a conserved DNA binding site. GT-1-containing transcription factors bind specific nuclear promoter sequences [61], making their presence in plant RNase J somewhat surprising. However, the structural conservation and retention of key residues hint that the GT-1 domain is functional in the context of RNase J.

The function of the GT-1 domain in plant RNase J remains enigmatic. While deletion of the GT-1 domain did not interfere with RNase J degradation activity in vitro when incubated with synthetic RNAs [52], it is more likely in vivo function would be related to sequence specificity, interaction with a PPR protein, and/or dimerization, which have not yet been rigorously tested. These possibilities are illustrated in Figure 3. For example, in mRNA 5′-end processing, the GT-1 domain could direct RNase J to certain locations on the RNA by direct sequence-specific binding or by binding to a sequence-specific cofactor (Figure 3 panel A). In the process of removing antisense transcripts, the GT-1 domain could influence RNase J target preference through its DNA, RNA, or protein-binding properties (Figure 3 panel B and see below).

## 5. Consequences of Removing or Down-Regulating RNase J in Plants

The only photosynthetic organisms in which an RNase J mutant phenotype has been studied are tobacco and *Arabidopsis*. *Arabidopsis* null mutants for RNase J are embryo-lethal, displaying albino ovules containing aborted embryos [63]. Further examination suggested that RNase J is required for the organization and functioning of the shoot apical meristems, cotyledons, and hypocotyls [64]. In addition, the transport and response of auxin was impaired [64]. Why the absence of RNase J activity results in embryo lethality is still obscure; however, the importance of plastid gene expression for embryo maturation in plants is well documented [65,66]. It is possible that simply impaired functioning of the chloroplast in general (see below), a specific function in the processing or degradation of a particular transcript, or another function that is not related to the ribonuclease activity is responsible for embryo lethality. In general, *At*RNase J is highly expressed in cells containing chloroplasts as well as in reproductive organs, and its expression is significantly light-dependent [64].

Because RNase J null mutants are embryo-lethal, virus-induced gene silencing (VIGS) was used to decrease RNase J abundance in tobacco and *Arabidopsis* [9,67]. The most striking effect of RNase J deficiency was massive accumulation of asRNAs, suggesting that the previously documented failure of chloroplast RNA polymerase to terminate efficiently [68] leads to symmetric transcription products that are normally eliminated by RNase J (Figure 4). This situation is exacerbated because chloroplast genomes are compact, with what generally appears to be a random distribution of genes on one strand versus another. In RNase J-down-expressed tissues, antisense–sense duplexes were readily detected and correlated with failure to associate with polysomes, chlorosis, and tissue death [9]. Therefore, in addition to its function in 5′-end processing, RNase J appears to play a major and essential role in chloroplast RNA quality control by eliminating long and otherwise abundant antisense transcripts. Open questions remain, however, as to whether a rapid elimination of antisense transcripts is required for the successful translation of the sense strand transcripts. It has long been known that transcription termination at the 3′ end of most genes is inefficient in chloroplasts, necessitating RNA maturation mechanisms to create defined 3′ termini [68]. Since chloroplast genomes are compact and, in most cases, have an apparently random distribution of genes on one strand versus another, there is a high potential for the accumulation of double-stranded molecules formed by sense and antisense transcripts. This situation is harmful for translation, therefore the antisense transcript would normally be rapidly eliminated [9]. The plant chloroplast RNase J has assumed the role of RNA surveillance, eliminating the antisense transcripts (Figure 4). Whether the GT-1 domain is important in this function, and more globally how RNase J differentiates between sense and antisense RNA to rapidly remove the second, is still obscure. A possible scenario imposing the plant specific GT-1 domain in this process is presented in Figure 3.

Analysis of VIGS-induced RNase J knockdown in plant tissue revealed that, in addition to its role in eliminating antisense transcripts, RNase J matures the 5′ ends of several transcripts, being guided or blocked by PPR proteins, as previously postulated [67,69] and consistent with in vitro analyses of its catalytic activity [9,52]. The 5′ maturation could occur by RNase J-catalyzed endonucleolytic cleavage followed by 5′→3′ exonuclease degradation until blocked by the corresponding PPR protein, generating the mature transcript 5′ end (Figure 5) [69]. The observed robust endoribonucleolytic and exonucleolytic activities of RNase J in vitro using a purified recombinant enzyme and a synthetic RNA supports that possibility. Otherwise, if RNase J is active in vivo exclusively as an exonuclease, the endonucleolytic cleavage could be performed by another endoribonuclease such as RNase E or RNase Z, followed by 5′→3′ exoribonucleolytic processive degradation by RNase J. Figure 5 illustrates the various modes of RNase J participation in chloroplast RNA 5′-end maturation, depending on whether the substrates are derived from intercistronic cleavage, nearby transcription initiation, or 3′ processing of an upstream tRNA.

## 6. Conclusions

Chloroplast RNA quality control is important for the maintenance of accurate gene expression, made necessary by relaxed transcription initiation and inefficient transcription termination. The evolutionarily conserved RNase J appears to play an essential role in this process by eliminating long antisense transcripts, in addition to its established role in 5′-end maturation. The structural and molecular mechanisms of these two processes remain understudied. Is *At*RNase J ribonucleolytic activity endo-, exo-, or both in vivo and how exactly does 5′-end processing occur, especially in terms of interactions with RBPs? How does the unique addition of the GT-1 domain to plant RNase J relate to its functions? The working hypotheses outlined in Figure 3, Figure 4 and Figure 5 point to future experimental approaches to address these questions.

## Figures and Tables

**Figure 1 plants-09-00334-f001:**
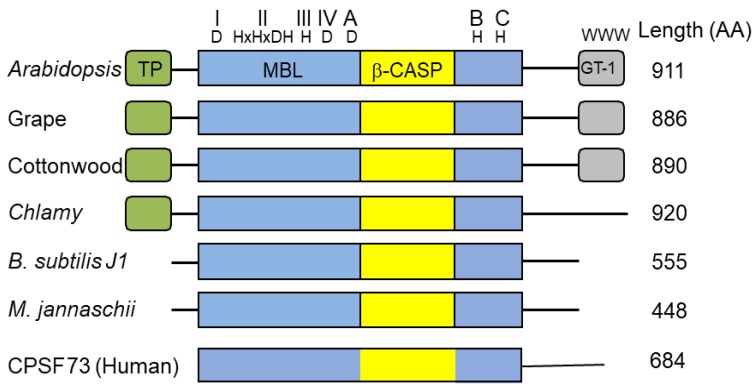
Domain comparison of several plant, bacterial, archaeal and human β-CASP metallo-β-lactamase (MBL) proteins. *Arabidopsis* RNase J (At5g63420) was used as a query to find homologous proteins. The domain structures from grape (*Vitis vinifera*; XM_002279762.1) and cottonwood (*Populus trichocarpa*; XM_002318086.1), representing plants, and from *Chlamydomonas reinhardtii* (GI: 187766729) *(Chlamy)*, the bacterium *Bacillus subtilis* (Q45493), and the archaea *Methanocaldococcus jannaschii* (Q58271) are presented, in comparison with human cleavage and polyadenylation specificity factor (CPSF)-73. The conserved motifs of MBL and β-CASP (I–IV; A–C) are indicated in blue and yellow, respectively, along with signature amino acid residues (above). Predicted chloroplast transit peptides (TP) are indicated in green. The plant C-terminus includes a region homologous to the GT-1 DNA-binding domain (grey). Its three conserved tryptophan residues are indicated.

**Figure 2 plants-09-00334-f002:**
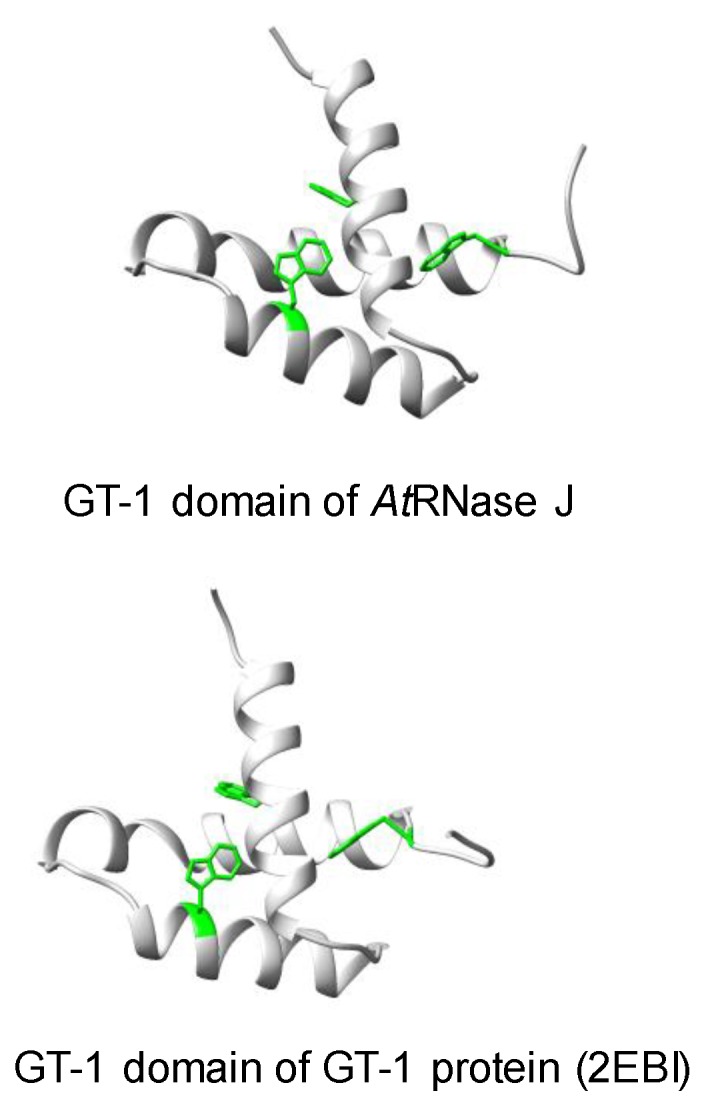
Comparative GT-1 domain structures. Top, three-helix model of amino acids (815–874) of the *Arabidopsis* RNase J (*At*RNase J) GT-1 domain, based on (bottom) PDB 2EBI (GT-1 transcription factor amino acids 81–152), built using the NMR-solved structure as a template [62]. The three conserved tryptophan residues are in green. (Acquired with copyright permission from [52]).

**Figure 3 plants-09-00334-f003:**
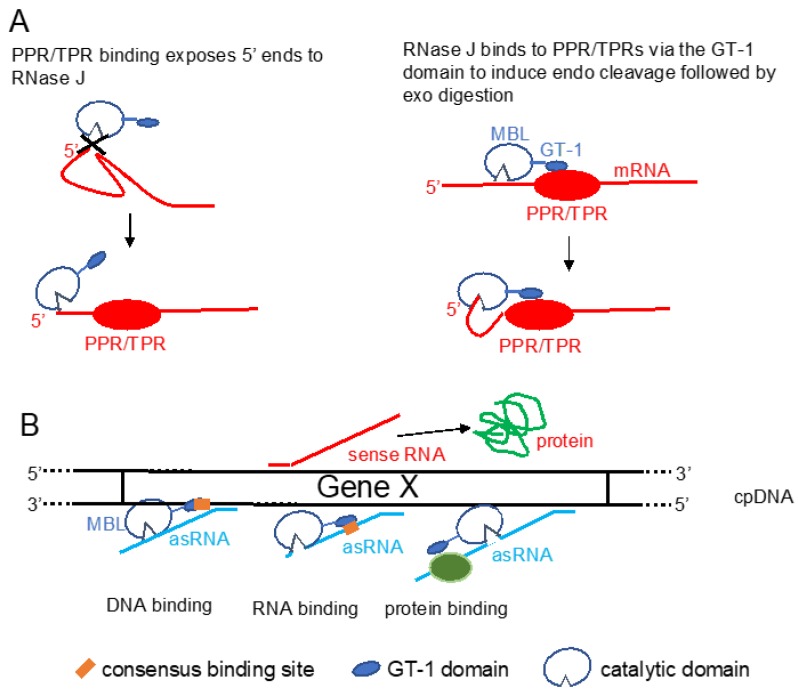
Putative functions of the GT-1 domain in models for RNase J modes of action. (**A**) Two scenarios for chloroplast (cp) 5′-end maturation by RNase J and the corresponding RNA-binding protein (RBP). **Left**, the RNA 5′-end structure prevents RNase J access. Binding of the RBP induces a structural change, exposing the 5′ end to digestion. **Right**, RNase J is recruited to the 5′ end by direct binding to the RBP, perhaps via the GT-1 domain. (**B**) Possible mechanisms of GT-1 domain-mediated recruitment of RNase J to targeted asRNAs by binding to a DNA site near the asRNA transcription start (**left**), to the asRNA itself (**center**), or to an RNA-binding cofactor (**right**). PPR/TPR: pentatricopeptide/tetratricopeptide repeat-containing proteins.

**Figure 4 plants-09-00334-f004:**
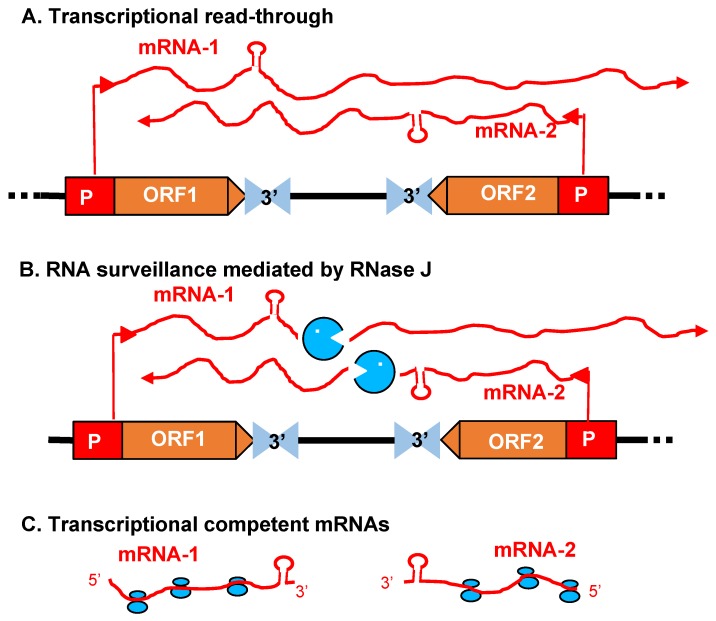
Model for antisense RNA (asRNA) surveillance by chloroplast RNase J. (**A**) 3′ UTRs of chloroplast genes inefficiently terminate transcription, resulting in read-through (mRNA-1 and mRNA-2). Where genes are convergently transcribed, even at a distance, asRNA may be synthesized. (**B**) These pre-mRNAs are first processed by an endonuclease, which could possibly be RNase J itself or another unidentified endonuclease. This creates substrates for the 5′→3′ exonucleolytic activity of RNase J. (**C**) By removing asRNA, RNase J allows the accumulation of single-stranded sense RNA that is translationally competent mRNA. (Acquired with copyright permission from [9]).

**Figure 5 plants-09-00334-f005:**
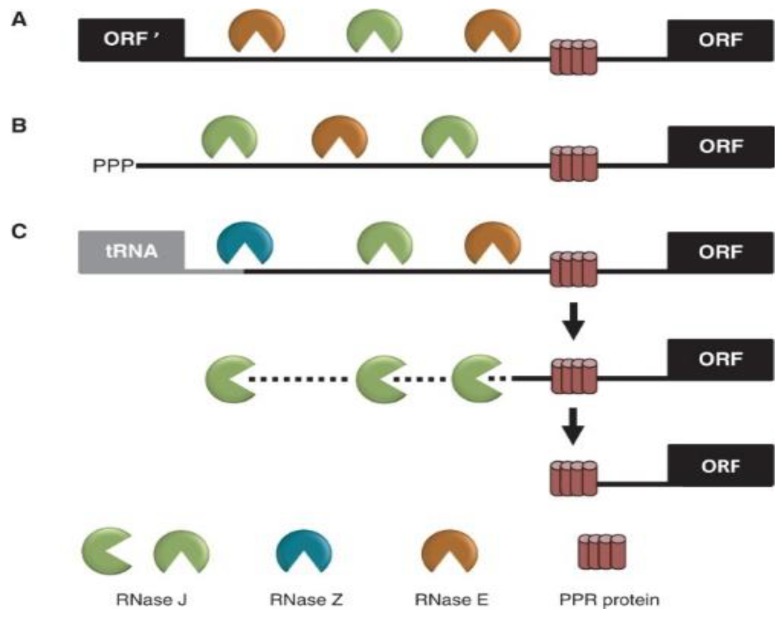
Model for the involvement of RNase J in processing chloroplast RNA 5′ ends defined by PPR proteins. Maturation of a generic mRNA (open reading frame, ORF) 5′ end is shown here. Precursor transcripts originate from polycistronic (**A**) and monocistronic (**B**) transcriptional units, as well as readthrough transcripts from upstream genes such as tRNAs (**C**). Processing is initiated by endonucleolytic cleavages by RNase J or RNase E within unstructured intergenic regions or, in the case of tRNAs, by RNase Z. The resultant 5′ ends are subsequently trimmed to their mature forms by the exonuclease activity of RNase J at PPR protein-bound sites. (Acquired with copyright permission from [67]).

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
