# Peer review of "Plant Ribonuclease J: An Essential Player in Maintaining Chloroplast RNA Quality Control for Gene Expression"

_plants, 2020, doi:10.3390/plants9030334_

Round 1
Reviewer 1 Report
This is an excellent review focusing on the description of documented and potential roles of chloroplast RNase J in the RNA quality control and gene expression. The authors provide information about the chloroplast RNA quality control and features of chloroplast gene expression in general, and then review recent studies on RNase J in various organisms. The review is very well written, easy to read and logically structured. The figures are mainly acquired from the author´s previous papers. Also Figure 1 and related Figure legend is practically identical to the Fig. 1A in Halpert et al. 2019, and I am wondering should a copyright permission be acquired for that as well, or at least to mention (with a reference to Halpert et al) that the figure is a modified version.
Minor comments:
Please clarify Figure 1: now the domain numbers are not clear and the text is overlapping with the boxes. Use scientific names in figure. If abbreviations are used, define them specifically in the legend (Does Chlamydomonas mean C. reinhardtii?)
Give reference for RNase J role in organization of shoot apical meristem and auxin responses (page 6, lines 230-231)
Rephrase the unclear title “Down expression of RNase J…” (e.g. Downregulated…)
Author Response
Please clarify Figure 1: now the domain numbers are not clear and the text is overlapping with the boxes. Use scientific names in figure. If abbreviations are used, define them specifically in the legend (Does Chlamydomonas mean C. reinhardtii?)
The reason that the domain numbers are not clear and that the text is overlapping with the boxes is because the figure was transferred from the power point figures file to the main manuscript word file after the submission and not by us. For that reason, the figures include also the Figure number. We asked the managing editor to take care in combining the revised version this time, as well as to let us review the final version again. The scientific names are now fully listed in the legend.
Give reference for RNase J role in organization of shoot apical meristem and auxin responses (page 6, lines 230-231)
The reference is listed: [65] (Chen, H.; Zou, W.; Zhao, J. Ribonuclease J is required for chloroplast and embryo development in Arabidopsis. J. Exp. Bot. 2015, 66, 2079–2091.)
Rephrase the unclear title “Down expression of RNase J…” (e.g. Downregulated…)
This title has been deleted and section 7 combined with section 6 (now labeled section 5), and in the process “Down expression” was changed to “down-regulation”
Reviewer 2 Report
This review by Hotto et al nicely summarized past knowledge and recent advances on role of RibonucleaseJ in chloroplast RNA quality control. The article is well written and nicely structured.
Here are few minor issues to fix before publication.
- What happens in terms of antisense RNA level if expression of RNaseJ is reduced in planta? Is it different in animals?
- What is the function of GT-1 domain in plant RNaseJ? What happens to its activity if this domain is deleted? Can animal version of RNaseJ complement plant RNaseJ mutant?
- Line80, what is genie transcript?
- Line 97, fix heading 4 title. Different prefix for CASP protein appears throughout this section.
Author Response
Here are few minor issues to fix before publication.
- What happens in terms of antisense RNA level if expression of RNaseJ is reduced in planta? Is it different in animals?
In section 5 (previously section 7) we refer to knockdown of RNase J using VIGS in tobacco and Arabidopsis. In both cases there was an increase in the accumulation of antisense RNA (see also Ref 9, Sharwood et al. 2011).
There is no RNase J in animals, perhaps since there is none in mitochondria. There is CPSF73 and other CPSF homologs in the nucleus and perhaps the cytoplasm. The only reports of down regulation of the CPSF73 expression we found resulted with the hampered 3’ end processing of Histones mRNA. However, this is not RNase J.
- What is the function of GT-1 domain in plant RNaseJ? What happens to its activity if this domain is deleted? Can animal version of RNaseJ complement plant RNaseJ mutant?
Excellent questions. We do not yet know the function of the GT-1 domain in plant RNase J. It is still an open question. Currently in vivo deletion mutants have not been produced or tested (to our knowledge), but deletion mutants did not seem to affect in vitro activity (see the last paragraph in section 5 and Ref 53, Halpert et al). Since there is no known animal version of RNase J (only CPSF), one cannot try complementing the chloroplast enzyme. However, it is certainly an interesting question of whether bacterial/archaeal/algal RNase J could complement plant RNase J.
- Line80, what is genie transcript?
This line says “genic transcript” which is referring to a transcript encoding a gene versus a non-coding transcript
- Line 97, fix heading 4 title. Different prefix for CASP protein appears throughout this section.
The symbols were changed from the original text. They have been changed back to β throughout the text, and hopefully will stay that way. Thank you for catching this.
Reviewer 3 Report
Very good, comprehensive review of the roles of RNAseJ in the chloroplast RNA quality control. This paper is well-written and structurized, references are relavant, hypotheses are interesting and thorough. This work falls into the scope of the Journal and will be useful for its readership.
I have two issues that should be addressed prior acception for publication:
- Figures need to be improved. (i) Their resolution is too low (structures in Fig 2, all details in Fig 3 and Fig 5); (ii) troubles with the text and different thickness of line (Fig 1); (iii) insufficient sharpness and contrast (image looks fuzzy).
- Conclusions (297-307) should be more concise and clear. If you meant your working hypotheses, please, repeat them in this section but do not refer to Figures, it seems strange.
Author Response
- Figures need to be improved. (i) Their resolution is too low (structures in Fig 2, all details in Fig 3 and Fig 5); (ii) troubles with the text and different thickness of line (Fig 1); (iii) insufficient sharpness and contrast (image looks fuzzy).
Agree. We will work with the editor to ensure that the final manuscript has clear images.
- Conclusions (297-307) should be more concise and clear. If you meant your working hypotheses, please, repeat them in this section but do not refer to Figures, it seems strange.
We have made some edits to the conclusions section to be more concise and clear. We did keep the reference to figures in the text as these figures illustrate the fundamental questions in our working hypothesis.